# Anise and Fennel Essential Oils and Their Combination as Natural and Safe Housefly Repellents

**DOI:** 10.3390/insects16010023

**Published:** 2024-12-29

**Authors:** Hataichanok Passara, Sirawut Sittichok, Cheepchanok Puwanard, Jirisuda Sinthusiri, Tanapoom Moungthipmalai, Kouhei Murata, Mayura Soonwera

**Affiliations:** 1Office of Administrative Interdisciplinary Program on Agricultural Technology, School of Agricultural Technology, King Mongkut’s Institute of Technology Ladkrabang, Ladkrabang, Bangkok 10520, Thailand; hataichanok.pa@kmitl.ac.th (H.P.); 63604010@kmitl.ac.th (C.P.); 2School of Agriculture and Cooperatives, Sukhothai Thammathirat Open University, Nonthaburi 11120, Thailand; sirawut.sit@stou.ac.th; 3Community Public Health Program, Faculty of Public and Environmental Health, Huachiew Chalermprakiet University, Samut Prakan 10540, Thailand; jiri_ja@yahoo.com; 4Department of Plant Production Technology, School of Agricultural Technology, King Mongkut’s Institute of Technology Ladkrabang, Ladkrabang, Bangkok 10520, Thailand; 64604012@kmitl.ac.th; 5School of Agriculture, Tokai University, Kumamoto 862-8652, Japan; kmurata@agri.u-tokai.ac.jp

**Keywords:** housefly, repellent, fennel essential oil, anise essential oil, synergistic effect, non-target species

## Abstract

Houseflies (*Musca domestica* L.) are an important medical and livestock pest that not only annoy animals and humans but can also transmit several human and animal diseases. Synthetic chemical repellents have been the first option for fly management, but most have negative impacts on humans, non-target organisms, and the environment generally. Plant essential oils are active against repelling insect pests and affecting their life cycles. Repellents from these oils are an important strategy as a new, effective, and environmentally friendly alternative for housefly management and reducing disease transmission. We investigated the housefly repellency and storage stability of fennel (*Foeniculum vulgare*) and anise (*Pimpinella anisum*) EOs and their combinations using the repellency of α-cypermethrin as a reference. Combinations (fennel + anise; 1:1) were the most effective and stable and more stable repellents than single EOs and α-cypermethrin. Most significantly, all single and combination essential oils were safe for two non-target species: guppies (*Poecilia reticulata*) and earthworms (*Eudrilus eugeniae*). The combinations make excellent, natural, and stable repellents for housefly management and are suitable to be developed into environmentally friendly products.

## 1. Introduction

Medical and livestock insect pests have a negative impact on human and animal welfare in several modes: they are disruptive, annoying, and vectors of diseases [1]. Among them, the housefly (*Musca domestica* L.) is a dominant pest that plays an important role as both a nuisance and a vector of several pathogens worldwide, including in Thailand [2,3]. It carries important diseases to humans and animals, including diarrhea, food-borne diseases, avian influenza, Turkey coronavirus, SARS-CoV-2, and COVID-19 [4,5]. Housefly management is difficult and complicated, especially in sensitive and high-density areas, due to its short life cycle, but flies readily reproduce on several types of organic matter, and, more importantly, they are resistant to several synthetic insecticides in several countries [6,7,8]. Moreover, several synthetic insecticides were found to be highly toxic to humans, animals, and non-target species, to have strong residual effects in the environment, and to disrupt the life cycle of natural enemies [9,10,11]. The over-application of synthetic insecticides did not impact human and animal well-being, but they affected environmental footprints that led to water, soil, human, and animal food contamination [12].

Consequently, one approach to the housefly problem is to improve innovative natural insecticides and repellents based on plant essential oils (EOs) as organic alternatives to synthetic insecticides [13,14]. They are strongly effective and characterized by eco-friendliness, biodegradability, low mammalian and non-target toxicity, and limited development of resistance, making them a suitable strategy for housefly management [15,16]. Plant EOs are active against houseflies as ovicides, oviposition deterrents, larvicides, pupicides, adulticides, and repellents [17,18,19,20,21]. Repellents play a very important role in disturbing and repelling flies and are safe in fly management to minimize synthetic insecticide residues [21,22].

Plant EO-based repellents are considered a safe and excellent choice for preventing flies in sensitive and epidemic areas that have not been used by synthetic insecticides, such as infant nurseries, kindergartens, nursing homes, cafeterias, postharvest areas, stored grain and fruit areas, and dairy farms [23,24,25].

Several reports presented EOs and their chemical compositions that effectively repelled houseflies. Table 1 lists several examples.

Against houseflies and other insect pests, EOs blocked vapors and damaged the sense of smell at antennal hairs [32,33]. EO vapors prevented flies and other insect pests from landing on and biting pets, animals, and humans [33,34]. EOs from lemongrass and star anise effectively repelled houseflies: they damaged antennae with abnormal, sunken, and twisted flagella and aristae [19,20].

Moreover, plant EOs are not only repellents but also safe for humans and are used in traditional medicine and additive foods, especially EOs from the Apiaceae family [35]. Among them, EOs from fennel and anise are the dominant ones that are effective as repellents and insecticides against several insect pests [36,37,38]. They are safe for non-target organisms, humans, and other animals [39,40,41].

Here, we investigated repellency and stability against houseflies of two single EOs and combinations of them—fennel and anise. These two EOs were selected as reported to be safe for mammals, non-target organisms, and ecosystems and have repellent and traditional medical properties [38,42]. The stability at several storage times, synergistic effects, and biosafety of single and combination EOs against a predator fish, guppies (*Poecilia latipinna)*, and earthworms (*Eudrilus eugeniae*), and scanning electron micrographs (SEM) of housefly antennae after exposure were checked. One limitation in EO-based repellent activity and stability is storage time [43]. The effectiveness of plant EO-based repellents depended on the volatility of EO constituents and the storage times. When the storage increased, repellent activity generally decreased [44], so this study was extended to 360 days.

The non-target guppy is a common aquatic predator in tropical areas, including Thailand [19,20]. Earthworms are also known as the “farmer’s friend” and are a soil-beneficial species for increasing soil nutrients and improving soil structure and ecosystems [45]; this species is widespread throughout tropical areas, including Asia and Thailand [45].

All single and combination EOs showed morphological damage in housefly antennae observed using optical and scanning electron microscopy. This repellency and stability showed they were a sustainable and safe alternative for housefly management, especially in sensitive and epidemic areas.

## 2. Materials and Methods

### 2.1. Essential Oils and Chemicals

The anise EO (CAS 8007-70-3) was purchased from Sigma-Aldrich Company Limited, Saint Louis, MO, USA. Fennel EO was purchased from the Nature In Bottle Corporation Company Limited, New Delhi, India. The two EOs and their combinations were used to prepare 70% (*v*/*v*) stock solutions in ethanol (purchased from Siam Medical Care Company Limited, Bang Bon, Bangkok, Thailand). The 10% (*w*/*v*) EC α-cypermethrin (Dethroid 10^®^), purchased from T.S. Inter Lab Limited Partnership Company, Bangkapi, Bangkok, Thailand, constituted the positive.

Anise has only one major component (*trans*-anethole, typically >88%, depending on the source). For the fennel, *trans*-anethole (80%) was also the main constituent, with minor constituents being 4-cymene (2.9%), fenchone (2.8%), and limonene (2.7%). In both cases, other constituents were less than 2% of the total [38,42,46].

### 2.2. Treatments

Based on our previous studies [19,20], the concentrations of two single EOs and their combination (anise EO: fennel EO; 1:1) treatments at concentrations of 0.74, 3.7, and 7.4 mL/m^3^ were held at room temperature (27.0 ± 3.5 °C). We prepared filtered samples, kept them in a 100 mL screw-topped reagent bottle, and assessed them for repellency after various storage times. At 1, 30, 90, 180, and 360 days, they were tested for repellency by exposing flies to them. A control using α-cypermethrin was prepared at the same concentration and used in the same way.

### 2.3. Housefly Rearing

Housefly adults were originally from the Entomology Laboratory, School of Agricultural Technology, King Mongkut’s Institute of Technology (KMITL), Ladkrabang, Bangkok, Thailand. They were bred in the Medical Entomology Laboratory, KMITL, at 25.5 ± 2.5 °C, 76.5 ± 2.3% RH, and alternating 12 h light and dark periods. Food for the adults was a mixture—honey + milk + mineral water—0.5:0.5:0.9 ratio—following Soonwera et al. [20]. After 1–2 days, generation one (G_1_) from the female adults laid eggs on steamed mackerel; the eggs developed into larvae, pupae, and adults. Repellent assays used the 3-day-old adults [20,47].

### 2.4. Repellency Activity Assay

A dual application method [20] was used to evaluate repellent efficacy against adults under controlled laboratory conditions. Initially, two test cages (300 mm × 300 mm × 300 mm) were connected by a rectangular hole (105 mm × 105 mm) (Figure 1). The first test cage was used for treated flies, and the second test cage was used for untreated flies. The 25 adults were released into each cage. The concentrations of each treatment and each exposure time—0.74, 3.7, or 7.4 mL/m^3^—were used in the treated cage; on the other side, ethanol was used in the untreated cage.

The α-cypermethrin positive control was tested concurrently. The number of flies that landed for at least 5 min at 6 h after exposure to the treated and untreated cages was recorded. The houseflies might land and then leave or stay on the filter paper until the end of each time period. All treatments used five replicates. Repellent rates (R) for adult flies were computed from Equation (1) [20,47]:R% = (U − T)/(U + T) × 100(1)
where U is the total number of flies landing on the untreated cage, and T is the total number of flies landing on the treated cage.

The effective repellent index (ERI) was calculated from Equation (2) [20]:ERI = RC_50 cyper_/RC_50 treat_(2)
where RC_50 cyper_ is the concentration at which 50% of flies were repelled by α-cypermethrin after 6 h, and RC_50 treat_ is the concentration at which 50% of flies were repelled after exposure for 6 h.

ERI indicates relative effective repellency, with ERI < 1 signifying that treatment was less effective than α-cypermethrin, whereas ERI > 1 indicates that treatment was more effective than α-cypermethrin.

The decreasing repellent index (DRI) was calculated from Equation (3) [20]:DRI = RC_50 day 360/_RC_50 day 1_(3)
where RC_50 day 360_ is the concentration at which 50% of flies were repelled after 6 h exposure to samples stored for 360 days, and RC_50 day 1_ is the concentration at which 50% of flies were repelled after 6 h exposure to samples stored for one day.

The synergistic repellent index (SI) was the relative efficacy of combination EOs over a single EO at the same strength. SI was calculated using Equation (4) [20]:SI = sum RC_50 sing_/RC_50 com_(4)
where RC_50 sing_ is the 50% repellency concentration of each single EO and RC_50 com_ is the 50% repellency concentration of the EO combination.

SI indicates relative synergy, with SI > 1 implying a positive synergy, whereas SI < 1 signifies a negative one.

### 2.5. Antennal Structural Changes from Optical and Scanning Electron Microscopy (SEM)

After the repellent bioassay, abnormal external and internal changes in antennae of treated versus untreated flies were observed by a stereomicroscope (Nikon^®^ Model C-PS, Nikon cell innovation Co., Ltd, Tokyo, Japan) and photographed with a digital camera (Nikon^®^ DS-Fi2, Nikon cell innovation Co., Ltd, Tokyo, Japan) at the Microscopy Centre, King Mongkut’s Institute of Technology Ladkrabang, Bangkok, Thailand [19,20]. Scanning electron micrographs were captured at the Chulalongkorn University, Scientific and Technological Research Equipment Centre, Pathumwan, Bangkok, Thailand [19,21].

SEM micrographs of fly heads—to view external ultrastructural changes in fly antennae following single and combination EO treatments versus controls. After 24 h, the heads of treated flies were cut off and placed in 70% ethanol for 30 min, then thoroughly washed with the same solution. Afterward, the head was postfixed in 95% (*v*/*v*) ethanol for 90 min and dehydrated with 100% ethanol.

Then, all head samples were dried with a CO_2_ critical point drier. Each dehydrated sample was mounted on aluminum stubs with double-sided adhesive tape and sputtered with gold. Micrographs were captured (JSM-6610LV SEM, JEOL Company Limited, Tokyo, Japan).

### 2.6. Safety Bioassay of Non-Target Species: Guppies and Earthworms

The toxicity of single and combination EOs was tested against non-target guppies following Soonwera et al. [48]. Guppies were purchased from an organic farm, Bigblue Inter Farm, Thailand (13°49′07.3″ N 100°46′46.1″ E). Fifty fish were kept in a plastic container (200 × 300 × 300 mm) containing 25 L of clean water at 28.8 ± 1.7 °C, 78 ± 2% RH, and 12 h alternating light and dark periods. The concentrations of each treatment were 100, 200, and 400 ppm. Ten adult fish were put in a plastic container (350 mm × 200 mm in height) containing 5 L of clean water. Five sets of fish were tested with α-cypermethrin. Guppy mortality was checked after 14 days post-treatment.

The toxicity assay against non-target earthworms followed OECD guidelines [49] and Jaitai [50]. The well-developed earthworms were obtained from an organic farm in Pak Chong, Thailand (14°42′45″ N 101°25′19″ E). Thirty earthworms were kept in a black plastic container (500 mm in diameter and 250 mm in height) containing 3 kg of artificial soil (natural fertilizer, cow manure, coconut coir, and organic soil—1:1:1:1 [50] under the same conditions as the guppy Bioassay). The concentration of each treatment was 100, 200, and 400 µL/kg of artificial soil. The 0.5 kg of wet artificial soil mixed with each treatment (pH 6.5–7.0; 65% soil moisture) was put in a black plastic container (200 mm in diameter and 180 mm in height), and ten earthworms were added. Each treatment was tested five times simultaneously with α-cypermethrin. Earthworm mortalities were checked after 14 days post-treatment.

### 2.7. Statistical Analysis

Statistical analysis used IBM’s SPSS Statistical Software Package version 28 (Armonk, NY, USA). We used a completely randomized design (CRD) for these bioassays. The mean repellent rate of all treatments and mean mortalities of non-target bioassays were analyzed by one-way analysis of variance (ANOVA), and Tukey’s test (*p* < 0.05) was used to evaluate the mean differences across multiple treatment groups [51]. Repellency, i.e., the concentration of a substance that repelled 50% (RC_50_) of adults by 6 h of exposure, was determined by probit analysis. Simple regression assessed the repellent efficacy against adult flies using generalized linear models with a binomial distribution. A correlation coefficient, R^2^, was used to evaluate linearity [52].

## 3. Results

### 3.1. Repellent Activity

Figure 2 shows regressions for repellent activity versus stored time of single and combination EOs at doses of 0.74, 3.7, and 7.4 mL/m^3^ against houseflies. All regressions had R^2^ ≈ 1.0, signifying that stored time strongly affected repellent activity. All treatments were repelled effectively at day 1, and the rate decreased with stored times from 30 to 360 days. These regressions showed that high doses (7.4 mL/m^3^) of all treatments were more effective repellents than lower doses (3.4 and 0.74 mL/m^3^). Both single EOs (fennel and anise) were less effective (with a repellent rate between 49 and 95%) than combination EOs (fennel EO + anise EO, 1:1) with repellent rates from 80 to 100%. Among the single EOs, fennel was more effective (with rates from 57 to 95%) than anise (rates from 49 to 88%). At 0.74 mL/m^3^ for exposures up to 360 days, combination EOs showed higher rates of 80 to 84%, while both single EOs showed rates from 49 to 72%, and α-cypermethrin only had rates from 14 to 31% (Figure 2A). At doses between 3.7 and 7.4 mL/m^3^ (Figure 2B,C), combination EOs showed 100% repellency at day 1, and the repellency after a long storage period with the repellent rate after 360 days ranged between 93 and 96%. Both single EOs repelled only 60 to 95% at the same dose. In contrast, α-cypermethrin repelled only 20 to 58%. All treatments were more effective than α-cypermethrin over the whole experimental period, in which treatments were stored for 360 days and then tested for storage period.

Figure 3 shows the repellent activity against houseflies measured by RC_50_, ERI, and DRI. Combination EOs were more effective (with RC_50_ from 0.4 to 0.8 mL/m^3^) than either single EO (with RC_50_ from 0.6 to 0.9 mL/m^3^). The combination EOs at day 1 showed the highest repellent activity with an RC_50_ of 0.4 mL/m^3^ and an ERI of 38, or over 38 times more effective than α-cypermethrin. From days 30 to 360, combination EOs also showed ERIs from 24 to 28 and were clearly more effective than α-cypermethrin. Fennel showed higher activities (RC_50_ from 0.6 to 0.9 mL/m^3^) than anise (RC_50_ from 0.7 to 0.9 mL/m^3^) or from 13 to 25 times more effective than α-cypermethrin, which had the lowest activity in all experiments, with RC_50_ from 9.2 to 25.2 mL/m^3^. Repellent activity in all treatments decreased over time. All repellents tested lost their DRIs by 1.3 to 2.0 times over 360 days.

Moreover, the repellency efficiency of combination EOs against houseflies over all periods was more than that of either single EO, with an SI of 2.4 to 3.8. The highest synergy was achieved by combination EOs after 30 days with an SI of 3.8. The synergistic effect decreased slightly with longer times, with SI down to 2.4 (Figure 4).

### 3.2. Morphological Changes

After 6 h of exposure to single and combination EOs, morphological alterations and abnormal antennae were recorded by optical and scanning electron micrographs (SEM)—see Figure 5A–G. The flies treated with fennel EO (C,D), anise EO (E,F), and anise + fennel EOs (G,H) showed remarkable shape aberrations and morphological damage of antenna with twisted scape and pedicel, distorted and wrinkled flagellum, lost setae, and distorted arista. The antennae of untreated controls showed a normal structure with scape, pedicel, and flagella (A,B).

### 3.3. Toxicity to Guppies and Earthworms

Toxicity of single EOs and combination EOs at 100 to 400 µL/L doses was assessed based on adult mortality after 14-day exposure (Table 2). All EOs were not toxic for guppies, whereas α-cypermethrin had the strongest toxicity, with up to 100% mortality at 400 µL/L. However, 100–200 µL/L doses were also toxic, with 70 to 98% mortality. Similarly, all EOs were not toxic for earthworms. In contrast, α-cypermethrin at 400 µL/kg was toxic to earthworms, with a 100% mortality (Table 3).

## 4. Discussion

The current global trend of growing human health concerns and insecticide resistance to synthetic insecticides in managing insect pests and insect vectors is increasing the pressure on scientists to find alternative natural pesticides and repellents [23,24,53,54]. Natural repellents from EOs represent a good option for developing new alternatives due to their widely presented repellent activity, which is biodegradable and non-mutagenic for mammals [20,24,25,26,27]. They are not only harmless for humans and some non-target species, but they are also highly effective repellents against houseflies [20,26,27]. They are also used as the primary prevention strategy to reduce the housefly populations in sensitive and outbreak areas [20,21,55,56]. It is significant that single EOs and combination EOs, including fennel and anise EOs, showed great potential as natural alternative repellents against houseflies and other vectors [26,27,56]. Many combination EOs from different plants showed synergies, where multiple EOs were better than single EOs, showing one strategy for improving repellent efficiency [20,24]. Moreover, using synergistic combinations in housefly management to reduce the total concentration or dose of the EOs led to higher repellent activity than that of a single EO [20,25]. Therefore, these combinations showed great potential as alternative repellents against adult flies.

In this study, a combination of EOs from fennel and anise (1:1) acted in synergy and showed stronger repellent activity against adult flies, based on several metrics, including low RC_50_, higher repellency rate, ERI, SI, and low DRI, when compared with fennel and anise alone at the same doses and exposures. These results were consistent with our previous work and several others showing high synergies reported for the combination of lemongrass + star anise (*Illicium verum*) (1:1), geranial + *trans*-anethole (1:1), and lemongrass + *trans*-anethole (1:1) used at 25 °C and 70% RH for 1 day that showed were 100% effective against housefly adults after 1 h exposure and repellency 34% more than a single EO [20]. Similarly, combinations of mentha + orange (1:1) (7:3), mentha + eucalyptus (7:3), and mentha + lemongrass (7:3) at a concentration of 0.025 µL/cm^3^ showed 100% repellency with 95% repellency concentration (RC_95_) around 0.01 mL/cm^3^ [27]. Phasomkusolsil et al. [44] also reported that a combination of (1:9) lemongrass + soybean oil against *Aedes aegypti* and *Anopheles dirus* mosquito adults showed high repellency with protection times up to 78 min. Benelli et al. [56] reported that a 1:2 mixture of ajwain (*Trachyspermum ammi*) + anise and a 1:1 mixture of wild celery (*Smyrnium olusatrum*) + anise showed strong larvicidal activity against filariasis vector larvae (*Culex quinquefasciatus*).

Storage times, volatility, and sensitivity to chemical reactivity limited the physical, biological, and quality of EO-based repellents [44,45]. In this study, our DRI of the combination showed the lowest (DRI = 0.8) and the highest repellent activity, although it was stored for up to 360 days. The physical properties of the combination, such as odor and color, were similar to those of the fresh combination on day 1. Similarly, Phasomkusolsil et al. [44] showed that lemongrass EO + soybean oil and citronella grass EO + soybean oil strongly repelled two mosquito vectors *(Ae*. *aegypti* and *An*. *dirus*) and were stable (no change in color of odor), stored at 25–30 °C for 180 days.

More importantly, our ERI showed that the combination of EOs from fennel + anise (1:1) was 28 times more potent than α-cypermethrin. Similarly, the combinations of EOs from lemongrass + star anise (1:1), lemongrass + nutmeg (*Myristica fragrans*) (1:1), and nutmeg + star anise (1:1) were 2 to 5 times stronger insecticides against housefly adults than α-cypermethrin [18,19,20,21]. We concluded that housefly adults had developed resistance to α-cypermethrin because it lost efficacy as both a repellent and an insecticide. We confirmed the findings of Abbas and Hafez [57], who showed that α-cypermethrin was less effective against adults and showed increased resistance ratios for housefly females from 46-fold at 5 generations to 470-fold, with similar changes for males. Similarly, Li et al. [58] reported that, in China, a field strain of housefly developed resistance to cypermethrin with a 153-fold resistance ratio. Zhang et al. [59] also reported long-term trends from 2003–2005 to 2021–2022, showing resistance to beta-cypermethrin increasing from 14% to 26%.

Optical and electron micrographs showed morphological alterations and abnormal antennae in housefly adults after treatment with single and combination EOs. The volatility and lipophilicity of the major component (*trans*-anethole) of both fennel and anise EOs were the main factors allowing the EOs to penetrate, interfere with, and disrupt the olfactory capabilities of the fly antennae [20,24,33,60,61], so the combination exhibited a high repellency for all tested exposures. The repellent receptors of insects differ; e.g., ticks detect repellents on the tarsi of the first legs, whereas houseflies, mosquitoes, and honeybees have repellent receptors on the antennae and palpi of the mouthpart [13,62]. The key compounds were attached to and blocked the sense of smell at olfactory or chemosensory receptors of mosquito antennae [13,34,62].

Similarly to Soonwera et al. [19,20], we observed similar significant morphological damage in housefly antennae and mouthparts following treatments with the EO combination of lemongrass + *trans*-anethole (1:1) and geranial + *trans*-anethole (1:1). Similarly, anise EO nanoemulsion destroyed the compound eyes and thoraces of a red flour beetle (*Tribolium castaneum*: Tenebrionidae) [60]. EOs from *Zanthoxylum limonell* + d-limonene (1:1) and star anise + *trans*-anethole (1:1) destroyed anal papillae and respiratory siphons of larvae and respiratory trumpets of pupae of two mosquitoes (*Ae*. *aegypti* and *Ae*. *albopictus*) [48]. Furthermore, Mougthipmalai et al. [62] reported morphological damage caused by d-limonene, geranial, and *trans*-anethole on exochorionic cuticles of eggs of these mosquitoes. Also, external changes in housefly eggs were caused by EOs from lemongrass, star anise, geranial, *trans*-anethole, lemongrass + *trans*-anethole (1:1), and star anise + geranial (1:1) [21].

In addition, EOs and their major components not only interfered with and disrupted the smell of fly antennae [20,62], but their major components showed several actions on flies and other pests [63,64]. The doses or concentrations of EOs for repelling insect pests and their mechanism of action are very important for improving new alternative repellents and safety for all mammals [20]. Anise EO was highly toxic to the nerve system of the two-spotted spider mite (*Tetranychus urticae*: Tetraychidae) because it inhibited acetylcholinesterase (AChE) and glutathione-S-transferase (GST). It also destroyed insect cuticles, which digested the protease enzyme [65]. Similarly, fennel EO and *trans*-anethole inhibited AChE in houseflies [66] and destroyed the cell cytoplasmic membrane of shigella bacteria (*Shigella dysenteriae*) [61]. Therefore, the fennel and anise EO combination was not only a repellent, but it also affected the physiology and killed houseflies.

However, EOs and their main compositions are considered eco-friendly and safe for pollinators and non-target predators, including fish and earthworms [19,20,21,48,62]. In this study, all EOs were not toxic for earthworms after 14 days. On the other hand, α-cypermethrin had the strongest toxicity to earthworms in the whole experiment. Benelli et al. [42] and Pavela [67] confirmed that fennel EO at 120 mg kg^−1^ and anise EO at 100 mg kg^−1^ were not toxic for earthworms, but α-cypermethrin at only 25 mg kg^−1^ had the strongest toxicity to earthworms. Similarly, EOs from Apiaceae (*Oliveria decumbens*) and three species of Lamiaceae (*Satureja sahendica*, *S*. *khuzestanica*, and *S*. *rechingeri*) at 200 mg kg^−1^ were not toxic for earthworms at 14 days after exposure, whereas α-cypermethrin at 0.1 mg kg^−1^ showed extreme toxicity with 100% mortality [68]. Similarly, all EOs were not toxic for guppies at 14 days after exposure. In contrast, 400 µL/L of α-cypermethrin led to 100% mortality at the same exposure time.

Similarly, EOs and their combinations, such as EOs from lemongrass, star anise, lemongrass + star anise (1:1), lemongrass + *trans*-anethole (1:1), star anise + geranial (1:1) at 1000–5000 ppm, were less toxic for guppies, with mortalities less than 10%, and LT_50_ ranged from 170 to 400 h after 10 days of exposure, whereas α-cypermethrin showed 98–100% mortality and LT_50_ ranged from 0.9 to 1.2 h [19,20,21]. Several monoterpenes and their combinations, such as geranial, *trans*-cinnamaldehyde, and geranial + *trans*-cinnamaldehyde (1:1) at 50–300 ppm, were less toxic to guppies, with mortalities less than 8%, while 1% α-cypermethrin caused 100% mortality [64]. EFSA [69] confirmed that α-cypermethrin had strong neurotoxic effects on mammals and humans, with acute oral LD_50_ of more than 150 mg/kg in rats and 15 mg/kg in beagles. However, doses of α-cypermethrin as low as 0.015 mg/kg in Wistar rats caused oxidative stress and DNA damage in blood plasma, liver, kidney, and brain tissue [70]. More importantly, cypermethrin was reported to cause long-term adverse effects on mammalian health, such as blood disorders, reproductive organ disorders, nervous system damage, and genetic disorders [69,70,71,72]. On the other hand, single EOs and their combinations of fennel and anise were safe and non-toxic to the tested guppies and earthworms. They were also non-toxic and thus safe for humans, pets, poultry, horses, and other domestic animals at acute oral LD_50_ of 3120 mg/kg in rats (fennel EOs) and 2090 mg/kg in rats (*trans*-anethole) [16,35,41,72]. They did not induce any genetic and DNA damage or cytotoxic effects on human cells [40,41]. They are permitted as food and feed additives in traditional and modern medicines and for other culinary and pharmacological purposes in several regions [40,41,73,74].

## 5. Conclusions

Our study suggests that the combination of EOs from fennel + anise (1:1) acted in synergy and remained effective after storage for 360 days at normal conditions. It is more effective than α-cypermethrin. It was thus suitable for further development into a natural repellent agent for housefly management. It was safe for non-target species, earthworms, and guppies. The repellency of this combination in field settings should be further studied. This combination could be used in alcoholic or aqueous solutions for managing housefly populations in households, farms, and other sensitive areas.

## Figures and Tables

**Figure 1 insects-16-00023-f001:**
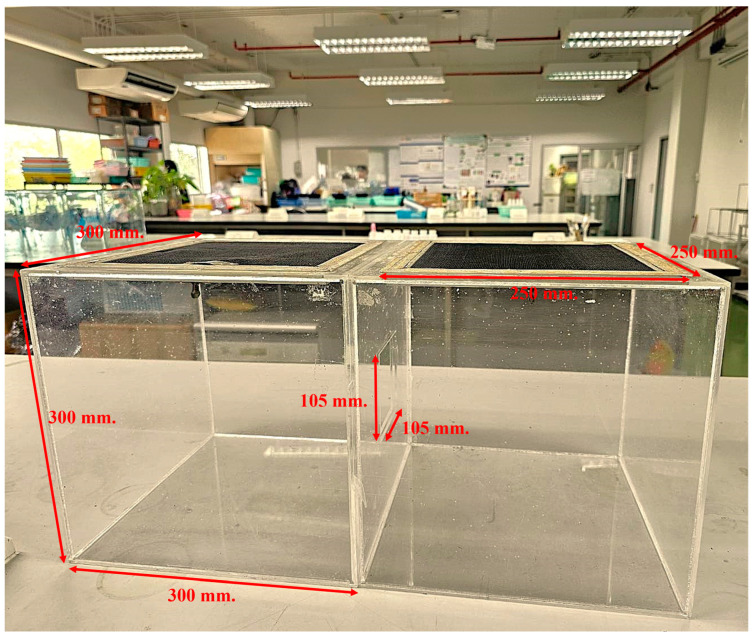
Repellency activity bioassay. The test cage (300 × 300 × 300 mm) was constructed from five plastic sheets, with one 250 × 250 mm screen at the top of the cage; the two compartments were connected by a 105 × 105 mm rectangular hole in the middle of the cage.

**Figure 2 insects-16-00023-f002:**
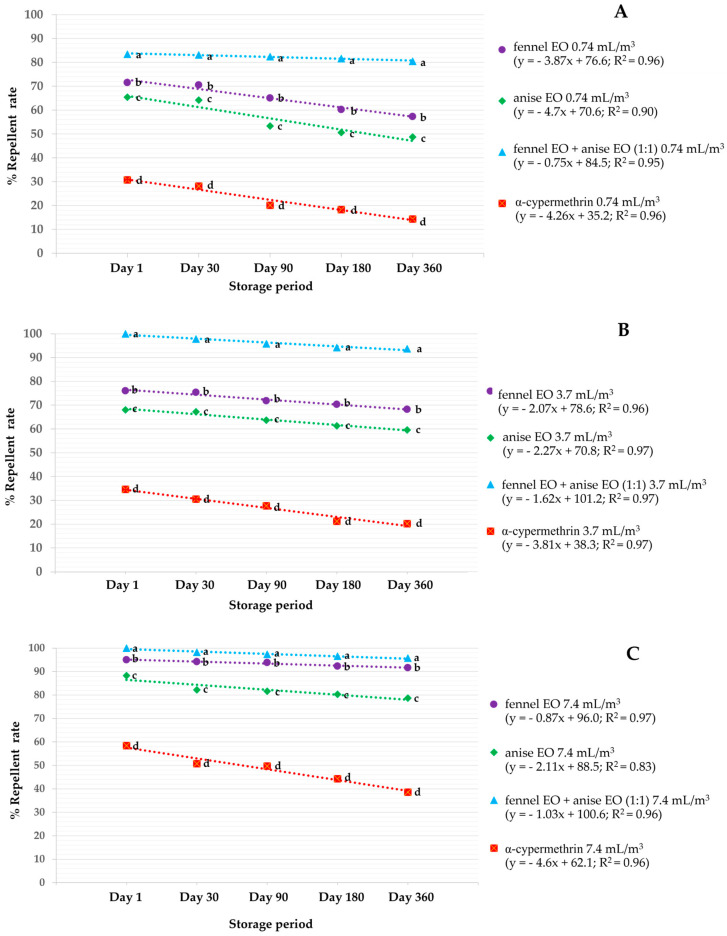
Repellent rates versus exposure of single and combination EOs against houseflies at doses of 0.74 mL/m^3^ (**A**), 3.7 mL/m^3^ (**B**), and 7.4 mL/m^3^ (**C**). For each test period, treatments marked by the same letter did not differ significantly (Tukey’s post hoc test *p* > 0.05).

**Figure 3 insects-16-00023-f003:**
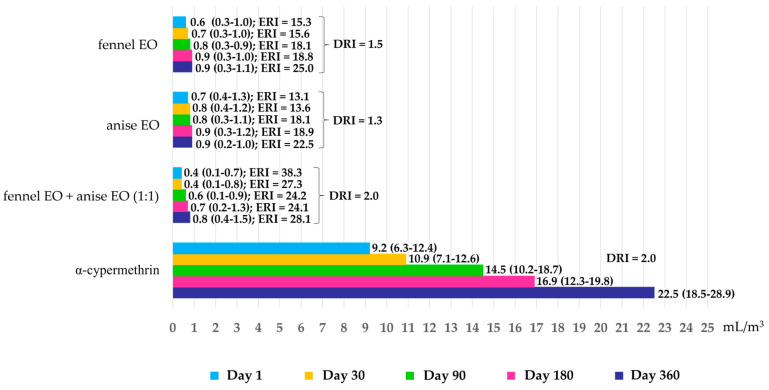
Repellency concentration of 50% (RC_50_) of single and combination EOs and α-cypermethrin against houseflies at exposure times varying from 1 to 360 days.

**Figure 4 insects-16-00023-f004:**
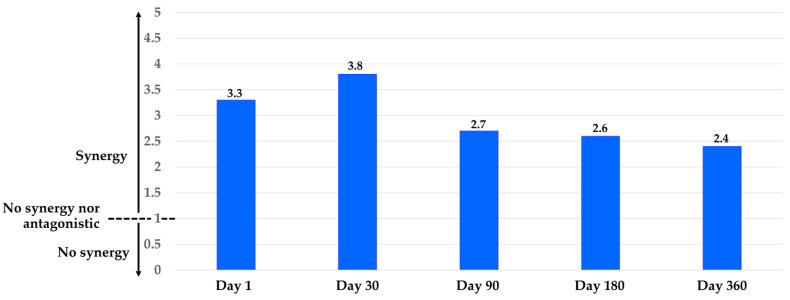
Synergistic Repellency Index (SI) of combination EOs against houseflies versus storage times from 1 to 360 days when compared to the single EO.

**Figure 5 insects-16-00023-f005:**
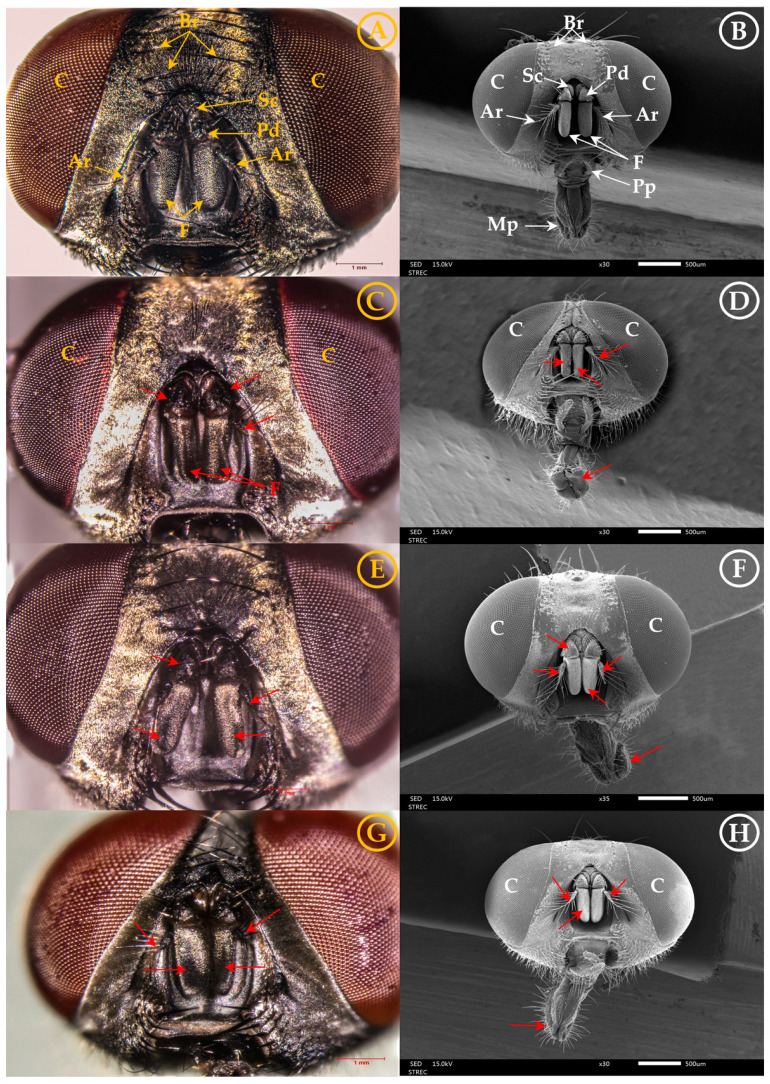
Optical and scanning electron micrographs showing the difference between untreated housefly antennae with optical and scanning electron microscopes (**A**,**B**), with the normal structure of scape (Sc), pedicel (Pd), flagellum (F), and arista (Ar) versus altered shapes (yellow and white arrow) and morphological damage of antennae with twisted scape and pedicel, distorted and wrinkled flagellum, lost setae, and distorted arista after exposure to fennel EO (**C**,**D**), anise EO (**E**,**F**), and anise + fennel EOs (**G**,**H**) (red arrow). Note: The housefly head has compound eyes (C), bristles (Br), antennae and mouth parts (Mp), and palpi (Pp).

**Table 1 insects-16-00023-t001:** Repellent activities of EOs and their compositions against adult houseflies.

EOs/Chemical Compositions	Concentrations	Repellent Rate (%)	Ref.
Geranial	1.0%	>90% at 2 h	[20]
*Trans*-anethole	1.0%	>90% at 1 h	[20]
Lemongrass (*Cymbopogon citratus*)	5.0%	>90% at 1 h	[20]
Geranial + *trans*-anethole (1:1)	2%	100% at 6 h	[20]
Peppermint (*Mentha piperita*)	70 µg/cm^2^	86% at 4 h	[26]
Blue gum (*Eucalyptus globulus*)	72 µg/cm^2^	76% at 4 h	[26]
Peppermint	0.01 µg/cm^2^	100% at 1 h	[27]
Peppermint + orange (*Citrus sinensis*) (1:1)	0.025 µg/cm^2^	100% at 1 h	[27]
Peppermint + lemongrass (1:1)	0.01 µg/cm^2^	100% at 1 h	[27]
*p*-anisaldehyde (anise: *Pimpinella anisum*, fennel: *Foeniculum vulgare*)	0.075%	60–78% at 4 h	[28]
Peppermint	1%	96.8% at 24 h	[29]
Cinnamon (*Cinnamomum verum*)	1%	77% at 24 h	[29]
Fennel	10%	100% at 5 d	[30]
Yellow oleander (*Thevetia peruviana*)	-	91.4% at 24 h	[31]
Neem (*Azadirachta indica*)	-	72.1% at 24 h	[31]
Eucalyptus (*E. camaldulensis*)	-	78.2% at 24 h	[31]

**Table 2 insects-16-00023-t002:** Toxicity of treatments and α-cypermethrin on non-target aquatic predator guppy at 14 days after testing.

Treatment	Conc. (µL/L)	Mortality Rate (%) ± SD				
		Storage Period (Day)				
		Day 1	Day 30	Day 90	Day 180	Day 360
Fennel EO	100	0 ± 0 ^c^	0 ± 0 ^c^	0 ± 0 ^c^	0 ± 0 ^c^	0 ± 0 ^c^
	200	0 ± 0 ^c^	0 ± 0 ^c^	0 ± 0 ^c^	0 ± 0 ^c^	0 ± 0 ^c^
	400	0 ± 0 ^c^	0 ± 0 ^c^	0 ± 0 ^c^	0 ± 0 ^c^	0 ± 0 ^c^
Anise EO	100	0 ± 0 ^c^	0 ± 0 ^c^	0 ± 0 ^c^	0 ± 0 ^c^	0 ± 0 ^c^
	200	0 ± 0 ^c^	0 ± 0 ^c^	0 ± 0 ^c^	0 ± 0 ^c^	0 ± 0 ^c^
	400	0 ± 0 ^c^	0 ± 0 ^c^	0 ± 0 ^c^	0 ± 0 ^c^	0 ± 0 ^c^
Fennel EO + Anise EO (1:1)	100	0 ± 0 ^c^	0 ± 0 ^c^	0 ± 0 ^c^	0 ± 0 ^c^	0 ± 0 ^c^
	200	0 ± 0 ^c^	0 ± 0 ^c^	0 ± 0 ^c^	0 ± 0 ^c^	0 ± 0 ^c^
	400	0 ± 0 ^c^	0 ± 0 ^c^	0 ± 0 ^c^	0 ± 0 ^c^	0 ± 0 ^c^
α-cypermethrin	100	80.0 ± 4.5 ^b^	76.0 ± 5.8 ^b^	75.0 ± 5.2 ^b^	75.0 ± 5.8 ^b^	70.0 ± 4.8 ^b^
	200	98.0 ± 5.6 ^a^	98.0 ± 6.8 ^b^	98.0 ± 5.6 ^b^	96.0 ± 6.5 ^b^	94.0 ± 5.5 ^b^
	400	100 ^a^	100 ^a^	100 ^a^	100 ^a^	100 ^a^
ANOVA *F*_0.05_, D*f*_total_, *p-value*		**, 59, *p < 0.01*	**, 59, *p < 0.01*	**, 59, *p < 0.01*	**, 59, *p < 0.01*	**, 59, *p* < 0.01

Mean percentage mortality rates in each column followed by the same letter are not significantly different (*p* > 0.05: Tukey’s test). ** Significantly different at *p* < 0.01.

**Table 3 insects-16-00023-t003:** Toxicity of treatments and α-cypermethrin to non-target earthworms at 14 days after testing.

Treatment	Conc. (µL/kg)	Mortality Rate (%) (Mean ± SD)				
		Storage Period (Day)				
		Day 1	Day 30	Day 90	Day 180	Day 360
Fennel EO	100	0 ± 0 ^c^	0 ± 0 ^c^	0 ± 0 ^c^	0 ± 0 ^c^	0 ± 0 ^c^
	200	0 ± 0 ^c^	0 ± 0 ^c^	0 ± 0 ^c^	0 ± 0 ^c^	0 ± 0 ^c^
	400	0 ± 0 ^c^	0 ± 0 ^c^	0 ± 0 ^c^	0 ± 0 ^c^	0 ± 0 ^c^
Anise EO	100	0 ± 0 ^c^	0 ± 0 ^c^	0 ± 0 ^c^	0 ± 0 ^c^	0 ± 0 ^c^
	200	0 ± 0 ^c^	0 ± 0 ^c^	0 ± 0 ^c^	0 ± 0 ^c^	0 ± 0 ^c^
	400	0 ± 0 ^c^	0 ± 0 ^c^	0 ± 0 ^c^	0 ± 0 ^c^	0 ± 0 ^c^
Fennel EO + Anise EO (1:1)	100	0 ± 0 ^c^	0 ± 0 ^c^	0 ± 0 ^c^	0 ± 0 ^c^	0 ± 0 ^c^
	200	0 ± 0 ^c^	0 ± 0 ^c^	0 ± 0 ^c^	0 ± 0 ^c^	0 ± 0 ^c^
	400	0 ± 0 ^c^	0 ± 0 ^c^	0 ± 0 ^c^	0 ± 0 ^c^	0 ± 0 ^c^
α-cypermethrin	100	88.0 ± 6.9 ^b^	87.0 ± 4.8 ^b^	87.0 ± 5.2 ^b^	85.0 ± 4.8 ^b^	84.0 ± 5.5 ^b^
	200	98.0 ± 5.6 ^a^	97.0 ± 6.8 ^a^	96.0 ± 6.6 ^a^	95.0 ± 5.5 ^a^	94.0 ± 4.6 ^a^
	400	100 ^a^	100 ^a^	100 ^a^	100 ^a^	100 ^a^
ANOVA *F*_0.05_, D*f*_total_, *p-value*		**, 59, *p < 0.01*	**, 59, *p < 0.01*	**, 59, *p < 0.01*	**, 59, *p < 0.01*	**, 59, *p* < 0.01

Mean percentage mortality rates in each column followed by the same letter are not significantly different (*p* > 0.05: Tukey’s test). ** Significantly different at *p* < 0.01.

## Data Availability

All relevant data are included in the article.

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
