# Peer review of "Anise and Fennel Essential Oils and Their Combination as Natural and Safe Housefly Repellents"

_insects, 2024, doi:10.3390/insects16010023_

Round 1

Reviewer 1 Report

Comments and Suggestions for Authors

Comments

1.       Please ensure that all species names are italicized the first time they are mentioned in the manuscript and references, with the author(s) and year of description included in parentheses immediately after the species name (without italics etc.

2.       Alpha-cypermethrin is a highly potent synthetic pyrethroid insecticide. This active ingredient is used in controlling various vector organisms such as cockroaches, house flies, and mosquitoes. According to standard WHO test methods, most insects (90%) exposed to surfaces treated with alpha-cypermethrin for 1 hour experience knockdown and die within a few hours. In this context, it is unclear how the repellent effect of alpha-cypermethrin on house flies was tested. Shouldn't insects that remain in contact with treated surfaces for up to 6 hours also experience knockdown, with a portion of them eventually dying? How did the authors interpret the inability of flies to fly, considering the lethal or knockdown properties of alpha-cypermethrin?

3.       It is stated that samples were taken over a 360-day period: Samples were taken after 1, 30, 90, 180, and 360 days. However, the method of sample collection is unclear from the information provided in the text. How exactly were the samples collected?

4.       The authors have stated in their previous publications (cited as references 19 and 20) that 1% alpha-cypermethrin exhibits highly lethal properties. How, then, can they claim in this publication that it demonstrates repellent rather than lethal effects over a 6-hour period?

5.       The description of the treatments in section 2.2 is unclear. When essential oils are dissolved in alcohol and sprayed into the environment (stated to be applied per cubic meter), droplets disperse and begin to evaporate under test conditions. How was contact between the insects and the essential oils or alpha-cypermethrin ensured? The test conditions are reported as 27°C ± 3°C. At these temperatures, wouldn’t the essential oils evaporate rapidly and lose their scent from the surfaces they contacted within a few days?

6.       Including photographs of the test environment where the repellent effect tests were conducted in the manuscript would help address readers' curiosity about how the tests were performed.

7.       Why was α-cypermethrin used in the repellent effect tests, given that it is a lethal active ingredient? Why wasn’t a commonly used repellent, such as DEET, used instead?

8.       The source of the houseflies (Musca domestica) is stated as the Entomology Laboratory, School of Agricultural Technology, King Mongkut’s Institute of Technology (KMITL), Ladkrabang, Bangkok, Thailand. However, the susceptibility of these flies to insecticides is not clarified, nor is it specified which generation of insects was used in the tests.

9.       Under what conditions were the environments and surfaces used in the tests stored for 360 days? Were factors such as liht exposure, humidity, etc., controlled and specified? What were the samples stored in at 27°C for 360 days, and why? Wouldn’t the essential oils tend to evaporate at 27°C or room temperature?

10.  What is the formulation type of α-cypermethrin (Dethroid 10®)? Is it EC, SC, or EW? Did spraying 3-7 ml of the product containing 10% alpha-cypermethrin into the test environment not exhibit lethal effects?

11.  What are the authors' comments on the inhalation toxicity of the essential oils and their combinations, which caused morphological abnormalities on the antennae of house flies, as well as their toxicity to the eyes of vertebrate organisms?

12.  Not all abbreviations used in the text are listed in the Abbreviations section.

13.  Please ensure that all references are formatted according to the journal’s style guidelines.

Comments on the Quality of English Language

good

Reviewer 2 Report

Comments and Suggestions for Authors

1.      What is the novelty of this research? How the study contributes to the field and addresses existing gaps?

2.      In Introduction section the authors claimed that they "investigated repellency and stability...". How the authors determined the stability of these essential oils? 

3.      Add a distinct section or paragraph at the end of the introduction that clearly enumerates the specific objectives of the study.

4.      Introduction section, line 97, author said: “One limitation in EOs-based repellent activity and stability is storage time”. Volatility of essential oils is a big common problem, how the authors solved it?

5.      Materials and method section, Repellency Activity Assay subsection, the authors stated that they evaluate repellent efficacy even after 360 days. It is not clear, if it is the same essential oils in the experiment? Does this mean that the essential oils are applied only once and the effects are monitored for a year? Please, be more precise.

6.      Considering that the authors are talking about a prolonged effect of essential oils, it is necessary to determine the release rates of the active components of the essential oils during the experiment.

7.      Please provide the chemical compositions of  these oils.

Round 2

Reviewer 1 Report

Comments and Suggestions for Authors

The authors have adequately addressed most of my comments.

Comments on the Quality of English Language

No comment

Author Response

We thank you very much, the reviewer, for making our paper more readable and useful for readers.

Reviewer 2 Report

Comments and Suggestions for Authors

The authors of the paper responded to all suggestions, improved the paper in scientific manner, and now the paper can be accepted for publication.

Author Response

(The authors gave the same response as above.)
